# RNA Modification-Related Genetic Variants in Genomic Loci Associated with Bone Mineral Density and Fracture

**DOI:** 10.3390/genes13101892

**Published:** 2022-10-18

**Authors:** Limin Han, Jingyun Wu, Mimi Wang, Zhentao Zhang, Dian Hua, Shufeng Lei, Xingbo Mo

**Affiliations:** 1Jiangsu Key Laboratory of Preventive and Translational Medicine for Geriatric Diseases, Department of Epidemiology, School of Public Health, Medical College, Soochow University, 199 Renai Road, Suzhou 215123, China; 2Center for Genetic Epidemiology and Genomics, School of Public Health, Soochow University, Suzhou 215123, China

**Keywords:** bone mineral density, RNA modification, genome-wide association study, gene expression

## Abstract

Genome-wide association studies (GWASs) have identified more than 500 loci for bone mineral density (BMD), but functional variants in these loci are less known. The aim of this study was to identify RNA modification-related SNPs (RNAm-SNPs) for BMD in GWAS loci. We evaluated the association of RNAm-SNPs with quantitative heel ultrasound BMD (eBMD) in 426,824 individuals, femoral neck (FN) and lumbar spine (LS) BMD in 32,961 individuals and fracture in ~1.2 million individuals. Furthermore, we performed functional enrichment, QTL and Mendelian randomization analyses to support the functionality of the identified RNAm-SNPs. We found 300 RNAm-SNPs significantly associated with BMD, including 249 m^6^A-, 28 m^1^A-, 3 m^5^C-, 7 m^7^G- and 13 A-to-I-related SNPs. m^6^A-SNPs in OP susceptibility genes, such as *WNT4, WLS, SPTBN1, SEM1, FUBP3*, *LRP5* and *JAG1*, were identified and functional enrichment for m^6^A-SNPs in the eBMD GWAS dataset was detected. eQTL signals were found for nearly half of the identified RNAm-SNPs, and the affected gene expression was associated with BMD and fracture. The RNAm-SNPs were also associated with the plasma levels of proteins in cytokine-cytokine receptor interaction, PI3K-Akt signaling, NF-kappa B signaling and MAPK signaling pathways. Moreover, the plasma levels of proteins (CCL19, COL1A1, CTSB, EFNA5, IL19, INSR, KDR, LIFR, MET and PLXNB2) in these pathways were found to be associated with eBMD in Mendelian randomization analysis. This study identified functional variants and potential causal genes for BMD and fracture in GWAS loci and suggested that RNA modification may play an important role in osteoporosis.

## 1. Introduction

Osteoporosis (OP) is a chronic disease characterized by decreased bone mass and damage to bone microstructures, which increases the risk of osteoporotic fracture. Due to the aging population, OP will become a serious public health problem threatening the health of middle-aged and elderly people. Bone mineral density (BMD) is an important indicator for diagnosing OP and predicting the risk of osteoporotic fracture. Calcium, vitamin D and antioxidants are beneficial to maintain BMD, while smoking and excessive alcohol consumption have adverse effects on BMD. Changes in physiological mechanisms, such as hormone levels, oxidative stress and cell apoptosis, also affect BMD levels.

BMD has a strong genetic component at all sites, with estimates of heritability ranging from 46% to 84% [1,2]. Genome-wide association study (GWAS) is a powerful tool to identify susceptibility genetic variants of complex diseases. In the past decade, large-scale GWASs have identified over 500 OP susceptibility loci [3,4,5,6]. However, exploring how these genetic variations affect BMD remains a challenge. Delineation of GWAS variants by distinguishing functional variants from other variants should help with the translation of GWAS signals into causal mechanisms and clinical applications. One of the most commonly used strategies in previous studies was applying exome sequencing technologies to detect potential functional variations that can alter amino acid sequences [7]. In fact, functional genetic variants may also be those influencing RNA–protein interactions [8] or changing the splicing sites of exonic splicing enhancers and silencers [9] through RNA editing [10].

RNA modifications that decorate the chemical and topological properties of ribose nucleotides, thereby executing their biological functions through post-transcriptional regulation, are important resources to increase our understanding of the mechanism of BMD loss. A wide range of RNA modifications, including m^6^A (N6-adenosine methylation), m^5^C (5-methylcytidin), A-to-I RNA editing, Nm (2′-O-ribose-methylation), Ψ (pseudouridine), m^7^G (N7-methylguanosine), and m^1^A (N1-adenosine methylation), have been studied. Dysregulation of RNA modifications affects a variety of biological processes, including cell proliferation, self-renewal procedures, and apoptosis.

In recent years, researchers have shown that genetic variants impact all types of RNA modifications by changing the nucleotides at which methylation occurs or RNA sequences around the target sites. Genetic variants may affect the regulation of gene expression by disturbing RNA modifications and, therefore, the RNA modification-related SNPs (RNAm-SNPs) may be important functional variants affecting BMD levels. In our previous study, we showed that m^6^A-SNPs may be among the functional variants for BMD [11]. Currently, the relationship between RNAm-SNPs and BMD is still less known. Annotating the functional impacts of RNAm-SNPs on BMD may be a valuable strategy to decipher its pathogenesis. Luo et al. developed a database called RMVar (http://rmvar.renlab.org accessed on 3 September 2020) to host genome-wide RNAm-SNPs. This database is helpful for functional studies on genetic variants affecting BMD.

This study attempted to evaluate the impact of nine types of RNAm-SNPs on BMD and identify potential functional genetic variants in gene loci associated with BMD in GWAS. The impacts of RNAm-SNPs on gene expression were evaluated in quantitative trait locus (QTL) studies, including mRNA expression QTL (eQTL) and plasma protein level QTL (pQTL), to support the functionality of the RNAm-SNPs. By applying Mendelian randomization (MR) analysis, gene expression and plasma protein levels involved in the regulation of RNAm-SNPs on BMD were identified (Appendix A).

## 2. Methods

### 2.1. Determination of RNAm-SNPs for BMD

In this study, we used summary-level data from two large-scale BMD GWASs to determine the potential functionally related RNAm-SNPs [5,6]. Estrada et al. performed a meta-analysis on lumbar spine (LS) and femoral neck (FN) BMD, including 32,961 individuals of European and east Asian ancestry. They identified 56 loci associated with BMD at the genome-wide significance level (*p* < 5.0 × 10^−8^) [6]. Morris et al. undertook a GWAS in 426,824 individuals from the UK Biobank and identified 518 genome-wide significance loci for BMD as estimated by quantitative ultrasound of the heel (eBMD). This study also identified 13 bone fracture loci, all of which were associated with eBMD, in ~1.2 million individuals [5]. The BMD GWAS datasets are all publicly available at the GEnetic Factors for OSteoporosis Consortium (GEFOS) website http://www.gefos.org/.accessed on 21 January 2022.

To identify the RNAm-SNPs in the large number of SNPs from the BMD GWAS, we obtained a set of RNAm-SNPs from the RMVar database (http://rmvar.renlab.org/download.html accessed on 21 January 2022). The RMVar database contains 1678,126 RNAm-SNPs for the nine types of RNA modifications (m^6^A, m^6^Am, m^1^A, 2′-O-Me, m^5^C, m^5^U, m^7^G, A-to-I and pseudouridine). These RNAm SNPs are divided into low (predicted), median and high confidence levels. The RMVar database characterized 31,076 RNA modification-related disease-associated variants by integrating the RNAm-SNPs with GWAS and ClinVar data, which directly provided clues to explore the relationship between RNA modification and disease of interest. Based on the RNAm-SNP information downloaded from the RMVar database, we annotated the SNPs in the BMD GWAS. The RNAm-SNPs associated with BMD were identified (significance level was set to *p* < 5.0 × 10^−8^) for further analysis.

### 2.2. Enrichment of RNAm-SNPs in the BMD GWAS Dataset

Among the BMD-associated SNPs, we determined if RNAm-SNPs were overrepresented compared to what would be expected by chance. We randomly sampled a set of non-RNAm-SNPs with the same number of RNAm-SNPs from the GWAS dataset for eBMD and calculated the proportion of SNPs with a *p* value < 5.0 × 10^−^^8^ in this set of non-RNAm-SNPs. Such an operation was repeated 1000 times. The distribution of the proportions resulting from this procedure was used as a background, and then the proportion of RNAm-SNPs with a *p* value < 5.0 × 10^−^^8^ was compared with this distribution, and a *p* value was reported.

In addition, we performed whole-genome analysis of RNAm SNPs by functional genome-wide association analysis (FGWAS) as a functional annotation of BMD. FGWAS is a program that uses association statistics calculated across the genome and incorporates functional genomic information into a GWAS to estimate the enrichment of GWAS hits in different annotation types [12]. Compared with existing methods for GWAS, FGWAS can substantially boost the detection power for discovering important genetic variants and the gene–environmental interactions influencing phenotypes and related functions. In addition, simulation studies show that FGWAS outperforms existing GWAS methods for searching sparse signals in an extremely large search space, while controlling for the family wise error rate.

### 2.3. eQTL Analysis for BMD-Associated RNAm-SNPs

RNA modification is important in gene expression regulation and mRNA stability and homeostasis, so RNAm-SNPs may be associated with mRNA levels. We are interested in whether the identified BMD-associated RNAm-SNPs affect mRNA expression levels and whether the affected gene expression was associated with BMD. We performed an eQTL analysis to identify the associations between RNAm-SNPs and gene expression levels in different types of cells and tissues. We obtained the eQTL information from a public database in the HaploReg browser (http://archive.broadinstitute.org/mammals/haploreg/haploreg.php accessed on 13 March 2022). The HaploReg browser developed by the Broad Institute can show the effect of SNPs on expression from eQTL studies, predict pathogenic variation and predict target genes and possible mechanisms of disease-related variation through systematic data mining. We focused on the association between RNAm-SNPs and genes located in *cis*-acting eQTLs.

Integration of GWAS data with eQTL studies is helpful to prioritize functionally relevant genes in GWAS-identified loci. Zhu et al. proposed an MR method named “summary data-based Mendelian randomization” (SMR) with this idea. In this study, we performed an SMR analysis to evaluate the associations between gene expression levels in five related tissues (whole blood, adipose tissue, skeletal muscle, liver and ovary) and eBMD and fracture by integrating eQTL data from the GTEx project [13] with BMD GWAS data described above [5]. A set of *cis*-eQTL summary data across the five human tissues from the GTEx project were downloaded from http://cnsgenomics.com/software/smr/#DataResource, accessed on 17 July 2020.

We ran SMR (version 0.712) with default parameters in a command-line program. The genotype data of HapMap r23 CEU were used as the reference panel to calculate the LD correlation for SMR analysis. The genome-wide significance threshold for the SMR analysis was set to 5.0 × 10^−6^. We further conducted the heterogeneity in dependent instruments (HEIDI) test to test the ‘no horizontal pleiotropy’ assumption, the basic assumption of the MR study. *p*_HEIDI_ > 0.05 indicated that there was one single SNP affecting BMD and gene expression.

### 2.4. pQTL Analysis for the BMD-Associated RNAm-SNPs

We carried out pQTL analysis for the identified RNAm-SNPs to search for plasma proteins related to BMD. Through pQTL analysis, we identified the proteins associated with BMD (pleotropic association) to further understand the physiopathology of BMD loss. The associations between RNAm-SNPs and plasma protein levels were searched in three pQTL studies. The first pQTL study was a large-scale proteomics GWAS that quantified 539 associations between protein levels and gene variants in a German cohort and replicated over half of them in an Arab and Asian cohort [14]. The summary data are available in the pGWAS Server database (http://metabolomics.helmholtz-muenchen.de/pgwas/index.php?task=download accessed on 13 December 2018). The second pQTL study characterized the genetic architecture of the human plasma proteome in 3301 participants aged 18 years or older who were recruited from two cohorts in England. Briefly, this GWAS tested the association between 10.6 million SNPs and 2994 plasma proteins [15]. The summary data are publicly available (http://www.phpc.cam.ac.uk/ceu/proteins/, accessed on 15 January 2020). The third pQTL study was a proteomics GWAS that measured 83 plasma proteins in a cohort of 3394 subjects [16]. The summary SNP data can be downloaded from https://zenodo.org/record/264128#.X3u5ga95uUk, accessed on 6 October 2020. In addition, we looked for associations between RNAm-SNPs and cytokines concentrations. The cytokine study studied the genetic basis for plasma levels of 41 cytokines in 8293 Finns [17]. The summary statistics of this study are available at http://www.computationalmedicine.fi/data#Cytokine_GWAS, accessed on 3 June 2019.

### 2.5. MR Analysis of Proteins

Horizontal pleiotropy is pervasive in MR analysis and can distort MR tests, leading to inaccurate causal estimates, loss of statistical power, and potential false positive causal relationships. To further assess whether there were potential causal effects between proteins identified by pQTL analysis and BMD, we performed weighted median, inverse-variance weighted (IVW) MR, MR–Egger, and MR pleiotropy residual sum and outlier (MR-PRESSO) analyses. The IVW method is a weighted linear regression model that combines the ratio estimates from each IV in a meta-analysis model. The premise for using this method is that all genetic variations are valid instrumental variables [18]. To obtain a more reliable MR estimate, we conducted MR–Egger regression. The MR–Egger method was performed to account for potential pleiotropy of the genetic variants that would have influenced the outcome through pathways other than the exposure, as this could have caused bias to the analytical results [19]. The weighted median, IVW and MR–Egger analyses were performed by using the Mendelian Randomization R package. MR-PRESSO is a method that systematically detects and corrects horizontal pleiotropic outliers in MR testing through three steps: the MR-PRESSO global test, the MR-PRESSO outlier test and the MR-PRESSO distortion test. The outlying genetic variants were identified by applying this method [20]. The source code and documents for MR-PRESSO are available at https://github.com/rondolab/MR-PRESSO, accessed on 17 July 2018. The default parameters were used for the MR-PRESSO analysis.

The requisite data (i.e., SNP rs number, β, standard error, and *p* value) were extracted from each of the BMD GWASs and pQTL studies mentioned above and then merged by SNP to form a plain file with seven columns (i.e., SNP rs number, β for protein, standard error for protein, *p* value for protein, β for BMD, standard error for BMD and *p* value for BMD) for the MR analysis using the R language. We sorted out the pQTLs with *p* values less than 5 × 10^−4^ as potential instrumental variables. We harmonized the genetic association between the pQTL and BMD GWAS to ensure that they reflected the same effect allele. We then conducted LD clumping on these SNPs to obtain the independent pQTL (LD *r*^2^ < 0.001, within 10,000 kb) for each protein. LD clumping was done using the clump_data function provided by the TwoSampleMR R package with reference to the 1000 Genomes EUR population.

### 2.6. Functional Enrichment Analysis

KEGG (Kyoto Encyclopedia of Genes and Genomes) is a collection of databases dealing with genomes, diseases, biological pathways, drugs and chemical materials that can help us understand the functional interpretation of genes and their products as a whole network. DAVID (https://david.ncifcrf.gov/ accessed on 13 May 2022) is an online bioinformatics tool that can provide comprehensive biological function annotation information for large-scale genes and proteins. In this study, we used the DAVID online tool to gain insights into the functions of the potential causal proteins.

## 3. Results

### 3.1. BMD-Associated RNAm-SNPs

In the BMD GWAS datasets, a total of 300 RNAm-SNPs that were significantly associated with eBMD at *p* < 5.0 × 10^−8^ were identified, including 249 m^6^A-, 28 m^1^A-, 3 m^5^C-, 7 m^7^G- and 13 A-to-I-related SNPs (Table 1, Appendix A). These RNAm-SNPs mapped to 255 known genes, including 220 protein coding genes (263 RNAm-SNPs) and 35 noncoding genes. The 300 detected RNAm-SNPs located in 243 GWAS identified loci. Most of these loci each contain only one RNAm-SNP, and 43 loci each contain two or more RNAm-SNPs. The identified RNAm-SNPs are always not the top significant SNPs in the loci. Forty-one (13.7%) of the RNAm-SNPs were “functional gain” while 259 (86.3%) were “functional loss”. These RNAm-SNPs were divided into three categories: 126 (42.0%) were high confidence, 75 (25.0%) were medium confidence and 99 (33.0%) were low confidence. Among the 263 RNAm-SNPs located in protein coding genes, 76 (28.9%) were exonic, 53 (20.2%) were in the 3′-UTR, 14 (5.3%) were in the 5′-UTR and 120 (45.6%) were intronic.

We found 249 m^6^A-SNPs that were significantly (*p* < 5.0 × 10^−8^) associated with eBMD (Table 1, Figure 1 and Appendix A), 215 of which were located in protein coding genes. One hundred and fifty (60.2%) genome-wide significant eBMD-associated m^6^A-SNPs belonged to the high and medium confidence categories. The top signals were found in the *IDUA* gene, including associations between rs115790973 (*p* = 2.90 × 10^−119^) and rs6815946 (*p* = 2.50 × 10^−116^) and eBMD (Figure 1 and Figure 2). Importantly, m^6^A-SNPs in key OP susceptibility genes were identified, including *WNT4*, *WLS*, *SPTBN1*, *SEM1*, *FUBP3*, *LRP5* and *JAG1* (Figure 1 and Figure 3). The m^6^A-SNPs rs7536301 in *WNT4* (*p* = 1.50 × 10^−22^), rs12044635 in *WLS* (*p* = 1.20 × 10^−28^), rs80233229 in *JAG1* (*p* = 4.70 × 10^−38^), rs62470375 in *SEM1* (*p* = 8.80 × 10^−10^), rs10901225 in *FUBP3* (*p* = 2.20 × 10^−24^), rs73516825 in *LRP5* (*p* = 1.50 × 10^−14^) and rs2229503 in *SPTBN1* (*p* = 1.70 × 10^−50^) (Figure 4) were significantly associated with eBMD.

Significant associations between m^6^A-SNPs and fracture were found, including rs115790973 (*p* = 1.00 × 10^−11^) and rs6815946 (*p* = 3.80 × 10^−11^) in *IDUA* (Figure 2), rs1983490 in LINC02594 (*p* = 3.60 × 10^−13^), rs2342312 in *WHSC1L2P* (*p* = 4.40 × 10^−13^) and rs8079310 in *ATXN7L3* (*p* = 1.20 × 10^−10^). The m^6^A-SNP rs10896350 in *PPP6R3* was significantly associated with LS-BMD (*p* = 3.68 × 10^−11^); rs1983490 in LINC02594 was significantly associated with LS-BMD (*p* = 2.88 × 10^−8^) and FN-BMD (*p* = 6.63 × 10^−9^). In addition, nominal significant associations were found for the known OP susceptibility genes, including the association between rs7536301 in *WNT4* and LS-BMD (*p* = 8.52 × 10^−5^), the association between rs12044635 in *WLS* and FN-BMD (*p* = 1.94 × 10^−5^), the association between rs62470375 in *SEM1* and FN-BMD (*p* = 7.13 × 10^−5^), the association between rs80233229 in *JAG1* and LS-BMD (*p* = 1.77 × 10^−6^) and the associations between rs273605 in *DCDC1* and FN-BMD (*p* = 1.30 × 10^−6^) and LS-BMD (*p* = 3.16 × 10^−7^).

For m^1^A-SNPs, we identified 34 functional loss m^1^A-SNPs that were significantly associated with eBMD. These 34 m^1^A-SNPs belong to the high and medium confidence categories (Appendix A). The top signal was found for rs227584 in *HROB* (*p* = 1.70 × 10^−41^), followed by the association between rs643892 in *LRP5* and eBMD (*p* = 9.00 × 10^−31^). The m^1^A-SNP rs227584 in *HROB* was also significantly associated with fracture (*p* = 6.90 × 10^−11^). The association between rs643892 in *LRP5* and LS-BMD was marginally significant (*p* = 8.63 × 10^−5^).

Four functional loss m^7^G-SNPs belonging to the medium confidence category were significantly associated with eBMD (Appendix A). Six functional loss A-to-I-SNPs belonging to the high confidence category were significantly associated with eBMD (Appendix A). For m^5^C modification, three functional loss m^5^C-SNPs belonging to the high confidence category were significantly associated with eBMD, including rs2229503 in *SPTBN1* (*p* = 1.70 × 10^−50^), rs11247975 in *SPON2* (*p* = 1.60 × 10^−23^) and rs9986596 in *ZKSCAN4* (*p* = 9.60 × 10^−9^). The m^5^C-SNP rs2229503 in *SPTBN1* is a synonymous variant and is also a m^6^A-SNP; rs11247975 in *SPON2* and rs9986596 in *ZKSCAN4* are missense variants.

### 3.2. Enrichment of RNAm-SNPs in the BMD GWAS Dataset

The proportion of m^6^A-SNPs, m^1^A-SNPs, m^7^G-SNPs, A-to-I-SNPs and m^5^C-SNPs that have GWAS *p* values < 5.0 × 10^−8^ for eBMD was significantly greater than that of the non-RNAm-SNPs (Table 1). We could not test the remaining four types of RNA methylation because of the lack of data. With the FGWAS method, it was found that SNPs associated with (*p* < 5.0 × 10^−8^) eBMD were significantly enriched with m^6^A-SNPs (log2 enrichment of 2.17, 95% CI: [1.00, 2.84]). Functional enrichments for the other types of RNA methylation in the eBMD GWAS dataset were not detected by using FGWAS software.

### 3.3. Gene Expression Associated with BMD

To further clarify the possible functional mechanisms underlying the identified RNAm-SNPs in association with BMD, we investigated whether they were associated with gene expression. We identified 253 BMD-associated RNAm-SNPs that showed effects in different cells or tissues, and cis-eQTL signals with corresponding local genes were found for 119 RNAm-SNPs (47.0%). Some identified RNAm-SNPs affect the expression of key OP susceptibility genes (Appendix A). Rs2229503, which is an m^6^A-SNP and m^5^C-SNP of *SPTBN1*, affects the expression of *SPTBN1*. The m^1^A-SNP rs643892 in *LRP5* is associated with the expression of *LRP5*. Four m^6^A-SNPs at 4p16.3 are associated with the expression of *FGFRL1*. The m^1^A-SNP rs227584 in *HROB* and the m^6^A-SNP rs8079310 in *ATXN7L3* are associated with the expression of *HDAC5*. Three m^6^A-SNPs (rs76324150, rs4792891 and rs17650901) in MAPT and m^7^G-SNP rs7350928 and m^6^A-SNP rs17574425 in *KANSL1* are associated with the expression of *WNT3*. The m^6^A-SNP rs978572 in *RELCH* is associated with the expression of *TNFRSF11A*.

In SMR analysis of integration of BMD GWAS with eQTL data from the GTEx project, we detected significant associations between gene expression in five related tissue types (whole blood, adipose, skeletal muscle, liver and ovary) and eBMD and fracture. A total of 142 significant pleotropic associations were detected for 23 of the genes containing RNAm-SNPs (SMR *p* < 5.0 × 10^−6^) (Appendix A). The number of significant associations found in each tissue was 77 in whole blood, 37 in adipose tissue, 19 in skeletal muscle, 4 in liver and 4 in ovary. In SMR analysis, we found that the expression levels of some OP susceptibility genes were significantly associated with eBMD in the five related tissue types. The expression of *FGFRL1* was associated with eBMD (*p* = 1.61 × 10^−8^) and fracture (*p* = 1.24 × 10^−5^) in skeletal muscle (Figure 2); the expression levels of *SPTBN1* were associated with eBMD in whole blood and adipose tissue (*p* = 3.15 × 10^−16^ and 6.55 × 10^−10^, respectively) (Figure 4).

### 3.4. Plasma Proteins Related to the RNAm-SNPs

We attempted to find plasma proteins that were related to the identified RNAm-SNPs. We found 340 significant pQTL signals (*p* < 5.0 × 10^−5^) for 96 RNAm-SNPs that were significantly associated with eBMD (*p* < 5.0 × 10^−8^) (Appendix A). A total of 180 proteins were detected, and these proteins were enriched in specific KEGG pathways, such as cytokine-cytokine receptor interaction (*p* = 6.30 × 10^−4^), the PI3K-Akt signaling pathway (*p* = 8.20 × 10^−4^), the NF-kappa B signaling pathway (*p* = 1.30 × 10^−3^) and the MAPK signaling pathway (*p* = 2.20 × 10^−3^) (Figure 5). Most of the pQTL signals were in trans effect, while rs12660627, rs8898, rs6815946 and rs28379706 were associated with plasma levels of proteins encoded by their local genes (*CD109*, *CTSB*, *IDUA* and *PLXNB2*, respectively). Most of these signals were found in the INTERVAL study, and signals for proteins encoded by *CTSB*, *IL6ST*, *SELP*, *ICAM1* and *MBL2* were found in more than one pQTL study. The m^6^A-SNP rs739468 in *CACFD1* was associated with plasma levels of 45 proteins. Other SNPs, such as rs41302673, rs2517719, rs200991, rs35835721, rs9986596, rs36019691 and rs56405707, were associated with plasma levels of more than 10 proteins. The top signal was the association between rs2645429 and plasma levels of CTSB, followed by the association between rs739468 and plasma levels of SELE. We also found that RNAm-SNPs were significantly associated with proteins encoded by *COL1A1*, *DOCK9*, *IBSP*, *MSRA, PILRA* and *PTHLH*, which are known to be important in OP.

### 3.5. Proteins Causally Associated with BMD

pQTL analysis showed that RNAm-SNPs were associated with plasma protein levels. To support the functional role of RNAm-SNPs in BMD, we still need to demonstrate that the plasma proteins affected by the RNAm-SNPs were associated with BMD. We chose proteins for MR analysis from three aspects based on the findings of the pQTL analysis. First, CTSB, IL6ST, SELP, ICAM1 and MBL2 showed significant signals in more than one pQTL study. Second, CTF1, IL1R1, IL19, LIFR, IL1B, IL3RA, CCL4, CCL3, CCL1, TNFSF8, CCL19, IL6ST, CCL28, LY96, TLR4, ICAM1, C4B, C4A, IRF1, FLT4, INSR, FN1, FLT3LG, EFNA5, COL1A1, IBSP, AKT2, KDR, MET, FGFR3 and CACNA2D3, which were enriched in specific KEGG pathways, were considered to be important. Third, CD109, COL1A1, DOCK9, IBSP, IDUA, MSRA, PILRA, PLXNB2, PRSS3 and PTHLH were also considered because RNAm-SNPs located in these genes affect plasma levels of proteins encoded by the local genes or because of their well-known roles in OP. Therefore, we tested whether these 42 proteins were genetically associated with eBMD using several MR methods.

For the five proteins that showed significant signals in more than one pQTL study, significant associations of CTSB with eBMD were detected using data from both the KORA and INTERVAL studies (Table 2). By using data from the KORA study, the plasma level of CTSB was associated with eBMD in weighted medium (*p* = 3.51 × 10^−10^), IVW (*p* = 5.82 × 10^−3^) and MR–Egger (*p* = 4.17 × 10^−6^) analyses. We found that the plasma level of CTSB was associated with eBMD in IVW (*p* = 4.41 × 10^−2^) and MR–Egger (*p* = 6.09 × 10^−4^) analyses using data from the INTERVAL study. The plasma level of MBL2 was associated with eBMD in weighted medium (*p* = 9.86 × 10^−12^) and MR–Egger (*p* = 2.41 × 10^−3^) analyses using data from the INTERVAL study, but the association may be due to pleiotropic effects because significant associations were not detected in MR-PRESSO analysis.

For the 31 proteins that were enriched in KEGG pathways, the plasma levels of IL19, LIFR, CCL1, CCL19, INSR, EFNA5, KDR and MET were found to be associated with eBMD in the MR analyses. For the remaining proteins, associations between plasma levels of CD109, COL1A1, DOCK9, PILRA, PLXNB2 and eBMD were found. However, the associations between plasma levels of CD109 and PILRA and eBMD may be due to pleiotropic effects because significant associations were not detected in MR-PRESSO analysis. Causal evidence was strong for the associations between plasma levels of COL1A1, IL19, KDR, MET and PLXNB2 and eBMD because the associations passed all four MR analysis tests and the intercepts of MR–Egger analyses were not significant.

According to the MR analyses, the plasma levels of ten proteins (CCL19, COL1A1, CTSB, EFNA5, IL19, INSR, KDR, LIFR, MET and PLXNB2) were associated with eBMD. We performed a functional enrichment for these ten proteins and found that these proteins were enriched in KEGG pathways of the PI3K-Akt signaling pathway (COL1A1, INSR, KDR, EFNA5 and MET), the Rap1 signaling pathway (INSR, KDR, EFNA5 and MET), the Ras signaling pathway (INSR, KDR, EFNA5 and MET) and the MAPK signaling pathway (INSR, KDR, EFNA5 and MET) (Figure 5).

## 4. Discussion

This study examined the association between RNAm-SNPs and BMD and showed that many BMD-associated SNPs in important genes identified by GWAS were related to RNA modification types of m^6^A, m^1^A, m^5^C, m^7^G and A-to-I. The findings indicate that RNA modification may play a role in OP, as the enrichment analysis showed that GWAS signals were significantly enriched with m^6^A-SNPs. Some of the identified RNAm-SNPs were located in well-known OP susceptibility genes and pathways. These SNPs showed *cis*-acting eQTL effects in relevant tissues, and some of them were found to be associated with proteins that were enriched in specific pathways. Moreover, the affected gene expression and protein levels were found to be associated with BMD. The results suggest a relationship among genetic variants, gene expression and BMD, i.e., the RNAm-SNPs affect RNA modification, which controls mRNA expression, and the altered mRNA expression or protein levels result in abnormal BMD.

Great efforts have been made to explore the relationship between genetic variations and diseases. However, distinguishing pathogenic variants from a large number of genetic variants remains a challenge. Most of the identified variants are functionally neutral, and only a few can cause disease. Increasing evidence shows that mRNA modifications play a critical role in modulating biological processes such as gene expression [21], mRNA stability [22] and homeostasis [23]. Annotating the functional effects of gene variants on RNA modification may be a valuable strategy to decipher the pathogenic mechanism of diseases. It is possible to determine the causal variants by measuring RNAm-SNPs associated with BMD. In this study, we demonstrated that searching for SNPs with specific functions in BMD-associated loci was essential for a better understanding of GWAS signals. We annotated many RNAm-SNPs related to BMD and showed that RNAm-SNPs may be involved in the pathogenesis of OP. Many of the identified BMD-associated RNAm-SNPs were involved in critical BMD genes and pathways. Well-known BMD genes, such as *WNT4*, *WLS*, *SPTBN1*, *SEM1*, *FUBP3*, *LRP5* and *JAG1*, contain RNAm-SNPs. We found evidence to show that the RNAm-SNPs may have impacts on the expression of these modifiable genes at both the mRNA and protein levels, and the affected gene expression was associated with BMD. Our research demonstrated how to refine the associated signals from the RNAm-SNPs identified in the GWAS dataset. The findings indicated that RNA modification may play a role in OP, as these key BMD genes were affected by RNA modification. Thus far, it is not clear how RNA modification affects these genes and contributes to OP, and its underlying mechanisms need to be clarified.

Rs2229503 in *SPTBN1*, rs7536301 in *WNT4*, rs76324150, rs4792891 and rs17650901 in *MAPT*, rs7350928 and rs17574425 in *KANSL1*, and rs643892 and rs73516825 in *LRP5* are RNAm-SNPs significantly associated with eBMD. Based on the RMVar database, rs2229503 is a m^6^A- and m^5^C-related SNP in *SPTBN1.* According to the HaploReg database, rs2229503 shows a *cis*-acting eQTL effect on the *SPTBN1* gene. *SPTBN1* is a candidate causal gene for BMD. Several GWASs have identified the relationship between *SPTBN1* and BMD [5,6], and functional studies have also suggested its functional relevance to the regulation of bone mass, supporting *SPTBN1* as a promising gene in the field of OP [24,25,26,27]. Upregulation of *SPTBN1* inhibited STAT3 signaling, whereas STAT3 was reported to be negative for bone homeostasis. Zhang et al. demonstrated that STAT3-deficient mice were prone to developing OP. In this way, *SPTBN1* suppressed STAT3 signaling, which led to decreased BMD and bone loss. *SPTBN1* passed the SMR analysis, which indicated that the expression level of *SPTBN1* was associated with BMD. Rs643892 is the m^1^A RNAm-SNP of *LRP5*, a component of the Wnt signaling pathway. According to HaploReg datasets, rs643892 is an eQTL SNP, showing a *cis*-acting eQTL effect on the *LRP5* gene. There is increasing evidence regarding the key role of the *LRP5* gene in regulating bone metabolism. The Wnt pathway plays a key role in bone metabolism [28]. It influences the differentiation and function of osteoblasts and osteoclasts, and its dysregulation leads to various forms of inherited bone mass disorders. RNAm-SNPs in *WNT4*, *WNT2B* and *APC* were associated with eBMD. RNAm-SNPs in *MAPT* and *KANSL1* were associated with the expression levels of *WNT3*. In addition to these genes, the expression levels of many genes affected by the RNAm-SNPs were found to be associated with BMD in relevant tissues. These RNAm-SNPs may be noteworthy functional SNPs. Modification of these sites may affect gene expression and disturb bone metabolism.

RNAm-SNPs may also affect gene expression at the protein level associated with BMD. According to pQTL analysis, RNAm-SNPs are pQTLs in genes known to be important in OP, including *COL1A1*, *DOCK9*, *IBSP*, *MSRA*, *PILRA* and *PTHLH*. The identified proteins also pointed to biological pathways highly related to OP, e.g., cytokine-cytokine receptor interaction, the PI3K-Akt signaling pathway, the NF-kappa B signaling pathway and the MAPK signaling pathway [29,30]. Most importantly, these proteins were found to be causally associated with BMD in MR analysis. The MR analysis highlighted proteins in the PI3K-Akt signaling pathway (COL1A1, INSR, KDR, EFNA5 and MET), the Rap1 signaling pathway (INSR, KDR, EFNA5 and MET), the Ras signaling pathway (INSR, KDR, EFNA5 and MET) and the MAPK signaling pathway (INSR, KDR, EFNA5 and MET), that may be candidate targets for OP. The relationships between many of the identified proteins, such as COL1A1, INSR, KDR and MET, and bone metabolism have been widely studied. MR analysis determined the risk factors for OP, and the results suggest that genes in the PI3K-Akt signaling pathway, the Rap1 signaling pathway, the Ras signaling pathway and the MAPK signaling pathway play functional roles in OP. The findings also indicate that RNAm-SNPs and RNA modification may play roles in OP through specific pathways.

This study has some potential limitations. First, the m^6^A-SNP set was large, but data for other types of RNA modification were very rare, so few RNAm-SNPs of these types were identified. Second, the functional effects of RNAm-SNPs on BMD were not examined experimentally. Further experiments in OP-related cells are needed to test their function.

## 5. Conclusions

This study annotated many RNAm-SNPs in many BMD-associated loci in GWAS, and elucidated the relationship between the SNPs and BMD-associated gene expression and protein levels. The findings of this study increase our understanding of the associations identified in BMD GWAS. The results suggest that RNA modification may be involved in the pathogenesis of OP. The RNAm-SNPs in BMD loci were associated with gene expression, including mRNA levels and protein levels, and the gene expression was associated with BMD, indicating that these genes may be causal factors for OP. No previous study has shown the relationship between these kinds of RNA modification and BMD. Therefore, this study may add valuable clues for further understanding the functional mechanism underlying the development of hypertension.

## Figures and Tables

**Figure 1 genes-13-01892-f001:**
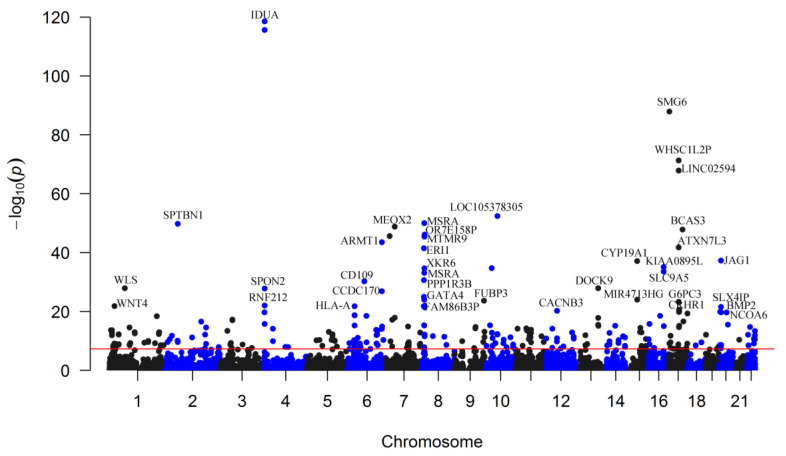
Genome-wide distribution of the identified eBMD-associated m^6^A-SNPs. The Manhattan plot shows the associations between m^6^A-SNPs and eBMD. The x-axis indicates chromosome positions. The y-axis indicates −log_10_*p* values of the associations. The red line indicates the genome-wide significance level of 5.0 × 10^−^^8^.

**Figure 2 genes-13-01892-f002:**
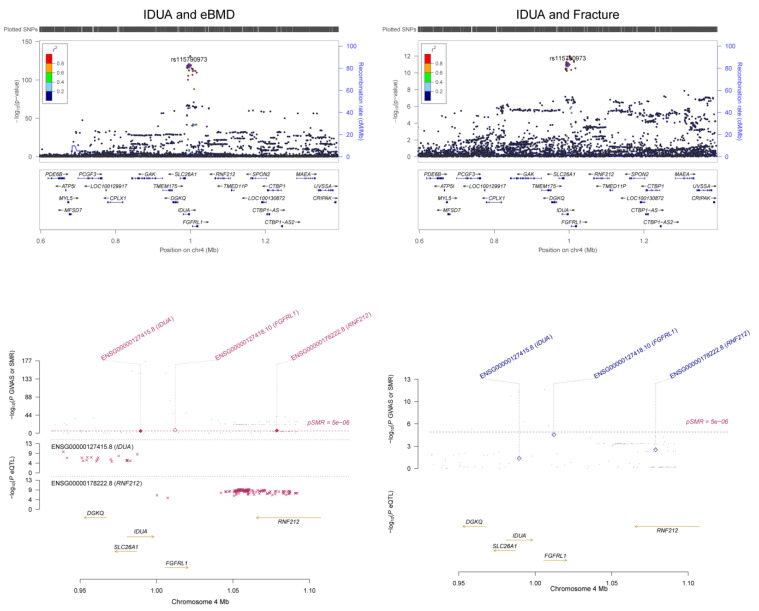
Association between the *IDUA* gene and eBMD. The m^6^A-SNP rs6815946 in the *IDUA* gene was associated with eBMD and fracture. The expression levels of the *FGFRL1* gene in skeletal muscle were associated with eBMD and fracture.

**Figure 3 genes-13-01892-f003:**
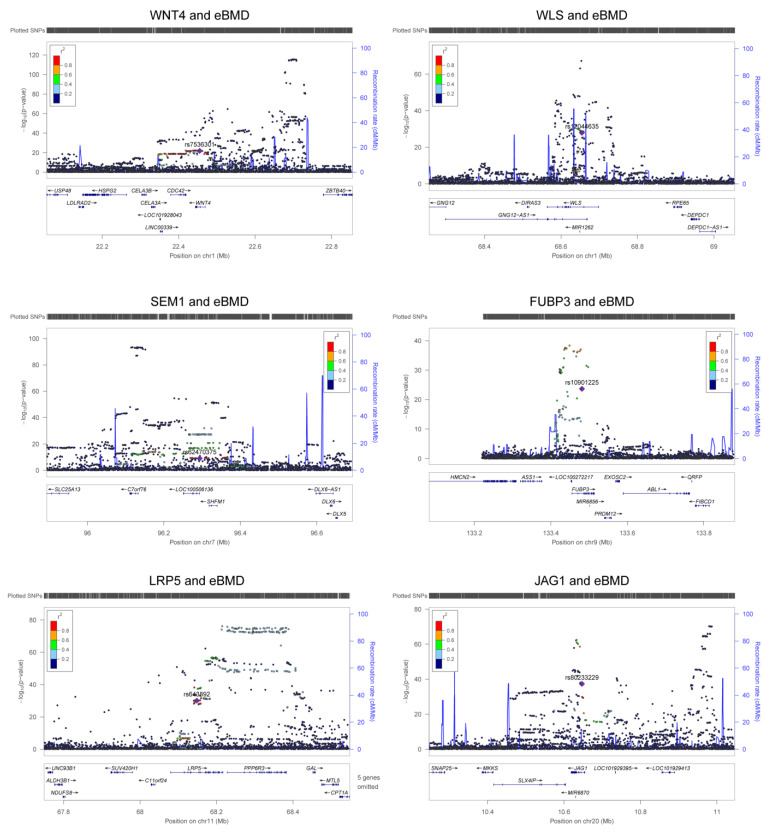
Association signals of m^6^A-SNPs with eBMD. The six regional association plots show the associations between m^6^A-SNPs in key OP susceptibility genes and eBMD. The m^6^A-SNPs in each gene locus are annotated in the plot.

**Figure 4 genes-13-01892-f004:**
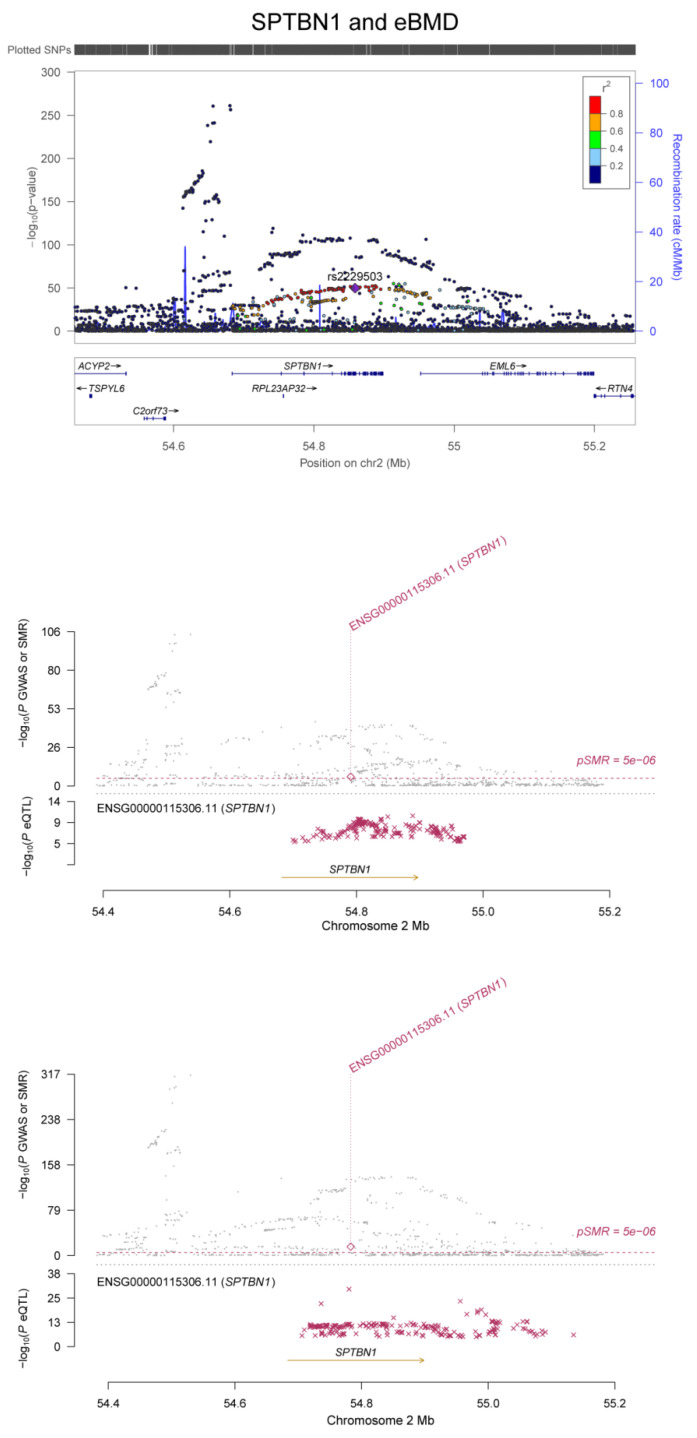
Association between the *SPTBN1* gene and eBMD. The m^6^A-SNP rs2229503 in the *SPTBN1* gene was associated with eBMD. The expression levels of the *SPTBN1* gene in adipose tissue and whole blood were associated with eBMD.

**Figure 5 genes-13-01892-f005:**
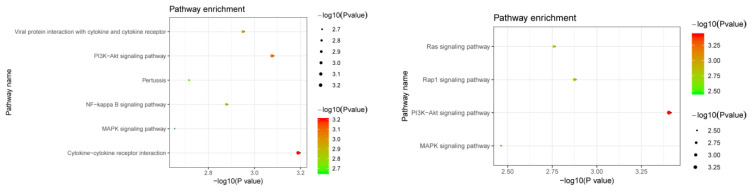
Pathway enrichment of the identified proteins. The upper panel shows the pathways for proteins affected by RNAm-SNPs identified in pQTL analysis; the lower panel shows the pathways for potential causal proteins identified in MR analysis.

**Table 1 genes-13-01892-t001:** Proportion of significant RNAm-SNPs in eBMD GWAS dataset.

RNAm	Total RNAm-SNPs Found in GWAS Dataset	RNAm-SNPs with *p* < 5.0 × 10^−8^ (%)	Simulated Proportion of Genome-Wide SNPs with *p* < 5.0 × 10^−8^ (95%CI)	*p* Value
m^6^A	18,082	249 (1.38%)	0.52–0.75%	0
m^1^A	1583	64 (4.02%)	0.25–1.07%	0
m^7^G	483	7 (1.45%)	0.41–0.83%	0
A-to-I	848	13 (1.53%)	0.47–0.83%	0
m^5^C	205	3 (1.46%)	0.49–0.98%	6.08 × 10^−145^
m^5^U	14	1 (7.14%)	-	-
m^6^Am	34	0	-	-
2′-O-Me	14	0	-	-
pseudouridine	12	0	-	-

**Table 2 genes-13-01892-t002:** Association between circulating protein levels and eBMD.

Proteins	Estimate ¶	Standard Error ¶	*p* Values				
			MR-PRESSO	IVW	Weighted Median	MR-Egger	Intercept
CTSB	−0.0311	0.0113	5.1 × 10^−2^	5.82 × 10^−3^	3.51 × 10^−10^	4.14 × 10^−6^	2.30 × 10^−3^
CCL1	−0.0134	0.0055	3.25 × 10^−2^	1.45 × 10^−2^	2.69 × 10^−2^	2.91 × 10^−1^	9.39 × 10^−1^
CCL19	0.0118	0.0037	6.11 × 10^−3^	1.82 × 10^−1^	2.03 × 10^−2^	5.80 × 10^−2^	1.28 × 10^−1^
CD109	−0.0190	0.0104	8.94 × 10^−2^	6.80 × 10^−2^	3.16 × 10^−2^	1.63 × 10^−2^	9.36 × 10^−2^
COL1A1	−0.0215	0.0060	4.06 × 10^−3^	2.27 × 10^−2^	5.34 × 10^−6^	5.96 × 10^−2^	8.10 × 10^−1^
CTSB	−0.0197	0.0098	6.02 × 10^−2^	4.41 × 10^−2^	1.49 × 10^−1^	6.09 × 10^−4^	7.29 × 10^−3^
DOCK9	0.0187	0.0060	7.98 × 10^−3^	1.75 × 10^−3^	7.40 × 10^−5^	3.96 × 10^−1^	7.15 × 10^−1^
EFNA5	0.0124	0.0049	2.73 × 10^−2^	4.03 × 10^−2^	6.06 × 10^−3^	3.07 × 10^−3^	3.78 × 10^−2^
IL19	−0.0133	0.0031	8.18 × 10^−4^	1.36 × 10^−3^	1.17 × 10^−6^	3.48 × 10^−7^	7.52 × 10^−1^
INSR	0.0249	0.0083	1.13 × 10^−2^	2.81 × 10^−3^	1.40 × 10^−3^	1.55 × 10^−7^	4.09 × 10^−4^
KDR	0.0143	0.0042	4.43 × 10^−3^	8.82 × 10^−5^	1.50 × 10^−2^	3.30 × 10^−3^	5.09 × 10^−1^
LIFR	0.0357	0.0155	4.94 × 10^−2^	2.07 × 10^−2^	5.27 × 10^−2^	8.53 × 10^−3^	6.99 × 10^−2^
MBL2	−0.0034	0.0047	4.82 × 10^−1^	2.61 × 10^−1^	9.86 × 10^−12^	2.41 × 10^−3^	5.12 × 10^−3^
MET	0.0385	0.0132	1.52 × 10^−2^	3.45 × 10^−3^	1.59 × 10^−2^	9.08 × 10^−3^	1.30 × 10^−1^
PILRA	−0.0103	0.0051	5.98 × 10^−2^	4.18 × 10^−2^	1.23 × 10^−2^	5.59 × 10^−1^	1.03 × 10^−1^
PLXNB2	−0.0156	0.0063	2.99 × 10^−2^	1.38 × 10^−2^	8.94 × 10^−9^	2.45 × 10^−2^	2.83 × 10^−1^

¶: The effect estimation was derived from the MR-PRESSO analysis.

## Data Availability

Data used in this study were provided in this paper.

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
