# Peer review of "RNA Modification-Related Genetic Variants in Genomic Loci Associated with Bone Mineral Density and Fracture"

_genes, 2022, doi:10.3390/genes13101892_

Round 1

Reviewer 1 Report

In this article, the authors aimed to attempted to evaluate the impact of nine types of RNAm-SNPs on BMD and identify potential functional genetic variants in gene loci associated with BMD in GWAS, and performed functional enrichment, QTL and Mendelian randomization analyses to support the functionality of the identified RNAm-SNPs.

The manuscript is of good quality, generally well-written and well structured. The topic is of major important for the current practice. Here I have several comments.

1.     In order for the reader to understand the flow of the study analysis, it seems necessary to present the flowchart of the analyzes performed in sections 2.3-2.6 at least as a supplementary figure.

2.     The tables (main and supplementary) were not included in the manuscript, making it difficult to follow the script of the results.

3.     The figures are difficult to visualize, it is suggested to improve the resolution

I would suggest the authors to improve the section of Introduction and detail the background of RNAm-SNPs related to this topic.

Author Response

  1. In order for the reader to understand the flow of the study analysis, it seems necessary to present the flowchart of the analyzes performed in sections 2.3-2.6 at least as a supplementary figure.

Response: Thank you for your thoughtful comments. According to your suggestion, we have made Figure S1 to show all stages in the data processing and the main results.

  1. The tables (main and supplementary) were not included in the manuscript, making it difficult to follow the script of the results.

Response: The tables were included in the revised manuscript. Thanks.

  1. The figures are difficult to visualize, it is suggested to improve the resolution

Response: The high resolution figures were submitted as separate files in the system. Thanks.

I would suggest the authors to improve the section of Introduction and detail the background of RNAm-SNPs related to this topic.

Response: In the Introduction section of the revised manuscript, we described the RNAm-SNPs in more detail (p 4, in red). Thanks.

Reviewer 2 Report

This study attempted to evaluate the impact of nine types of RNAm-SNPs on BMD  and identify potential functional genetic variants in gene loci associated with BMD in  GWAS. The impacts of RNAm-SNPs on gene expression were evaluated in quantitative  trait locus (QTL) studies, including mRNA expression QTL (eQTL) and plasma protein  level QTL (pQTL), to support the functionality of the RNAm-SNPs. By applying Mende lian randomization (MR) analysis, gene expression and plasma protein levels involved 80 in the regulation of RNAm-SNPs on BMD were identified. This study annotated many RNAm-SNPs in many BMD-associated loci in GWAS and elucidated the relationship between the SNPs and BMD-associated gene expression  and protein levels. The findings of this study increased our understanding of the associations identified in BMD GWAS. The results suggested that RNA modification may be  involved in the pathogenesis of OP. The RNAm-SNPs in BMD loci were associated with  gene expression, including mRNA levels and protein levels, and the gene expression  was associated with BMD, indicating that these genes may be causal factors for OP. No  previous study has shown the relationship between these kinds of RNA modification  and BMD. Therefore, this study may add valuable clues for further understanding the  functional mechanism underlying the development of hypertension.

The introduction is well written, with adequate citations. State the problem and clearly state the goal

The methodology is clearly described so that other authors are able to repeat the experiments. The results are complex and sometimes difficult to understand A table, highlighting the most striking data could be useful The discussion is correct in emphasizing the importance of functional SNPs. The absence of citations in the discussion is striking. This fact should be changed

Author Response

Response: Thank you for your thoughtful comments. Because the number of tables/figures were restricted, we listed the identified RNAm-SNPs and genes in supplementary tables. The important genes were highlighted in red in the supplementary tables. The identified proteins were highlighted in table 2. In this revision, we have cited more references in the Discussion section. Thank you very much for all of your thoughtful comments and suggestions. We believe that our manuscript has been greatly strengthened by this revision.
